# The BigBrainWarp toolbox for integration of BigBrain 3D histology with multimodal neuroimaging

Casey Paquola[1,2]*, Jessica Royer[1], Lindsay B Lewis[1], Claude Lepage[1], Tristan Glatard[3], Konrad Wagstyl[4], Jordan DeKraker[1,5], Paule-J Toussaint[1], Sofie L Valk[6,7], Louis Collins[1], Ali R Khan[8], Katrin Amunts[2], Alan C Evans[1], Timo Dickscheid[2], Boris Bernhardt[1]*

[1]McConnell Brain Imaging Centre, Montreal Neurological Institute and Hospital, McGill University, Montréal, Canada; [2]Institute of Neuroscience and Medicine (INM-1), Forschungszentrum Jülich, Jülich, Germany; [3]Department of Computer Science and Software Engineering, Concordia University, Montreal, Canada; [4]Wellcome Trust Centre for Neuroimaging, University College London, London, United Kingdom; [5]Brain and Mind Institute, University of Western Ontario, Ontario, Canada; [6]Otto Hahn Group Cognitive Neurogenetics, Max Planck Institute for Human Cognitive and Brain Sciences, Leipzig, Germany; [7]Institute of Neuroscience and Medicine (INM-7), Forschungszentrum Jülich, Jülich, Germany; [8]Department of Medical Biophysics, Schulich School of Medicine & Dentistry, University of Western Ontario, London, Canada

**Abstract** Neuroimaging stands to benefit from emerging ultrahigh-resolution 3D histological atlases of the human brain; the first of which is 'BigBrain'. Here, we review recent methodological advances for the integration of BigBrain with multi-modal neuroimaging and introduce a toolbox, 'BigBrainWarp', that combines these developments. The aim of BigBrainWarp is to simplify work-flows and support the adoption of best practices. This is accomplished with a simple wrapper function that allows users to easily map data between BigBrain and standard MRI spaces. The function automatically pulls specialised transformation procedures, based on ongoing research from a wide collaborative network of researchers. Additionally, the toolbox improves accessibility of histological information through dissemination of ready-to-use cytoarchitectural features. Finally, we demonstrate the utility of BigBrainWarp with three tutorials and discuss the potential of the toolbox to support multi-scale investigations of brain organisation.

*For correspondence:
casey.paquola@gmail.com (CP);
boris.bernhardt@mcgill.ca (BB)

Competing interest: The authors declare that no competing interests exist.

## Introduction

Understanding brain anatomy requires a multi-scale perspective. Regional variations in cell types and distributions underlie macro-scale patterns, whether they are reflective of functional dynamics, age, or disease states. For over 150 years (*von Gudden, 1886*), histological analysis of post mortem tissue has helped to reveal the microscopic architecture of the brain. Neuroanatomists observed a distinctive layered organisation of cells within the cortex (*Baillarger, 1840*), identified differences in the cellular composition (*Betz, 1874*), and developed principles of cortical organisation, including the definition of cortical types (*Meynert, 1867*) and areas (*Brodmann, 1908*; *Von Economo and Koskinas, 1925*). More recently, digitisation of post mortem tissue has allowed automated characterisation of cytoarchitecture and the definition of borders between areas (*Schleicher et al., 1999*).

Evidence has been provided that cortical organisation goes beyond a segregation into areas. For example, large-scale gradients that span areas and cytoarchitectonic heterogeneity within a cortical area have been reported (*Amunts and Zilles, 2015*; *Goulas et al., 2018*; *Wang, 2020*). Such progress became feasible through integration of classical techniques with computational methods, supporting more observer-independent evaluation of architectonic principles (*Amunts et al., 2020*; *Paquola et al., 2019*; *Schiffer et al., 2020*; *Spitzer et al., 2018*). This paves the way for novel investigations of the cellular landscape of the brain.

In vivo neuroimaging offers a complementary window into the structure and function of the brain. The non-invasive nature of magnetic resonance imaging (MRI) allows examination of large-scale, population-level variation, which is more limited in *post mortem* neuroanatomy. Human brain mapping research has furthermore established standard spaces, notably the MNI152 space for volumetric whole-brain analysis (*Fonov et al., 2011b*; *Fonov et al., 2009*; *Mazziotta et al., 2001a*; *Mazziotta et al., 2001b*) and 'fsaverage' and 'fs_LR' for surface-based cortical analyses (*Fischl et al., 1999*; *Van Essen et al., 2012*). Despite ongoing advances in attaining higher spatial resolution with higher field strength (*Deistung et al., 2013*; *Holdsworth et al., 2019*; *Sitek et al., 2019*; *Trampel et al., 2019*; *Turner and De Haan, 2017*), in vivo MRI researchers remain constrained by limited spatial resolution from making inferences on a cellular level. Establishing the relationship between macro-scale patterns and cellular architecture is crucial to substantiate physiological patterns observed with MRI and for further development of brain-inspired computational models.

BigBrain is a singular 3D volumetric reconstruction of a sliced and cell-body stained post mortem human brain (*Amunts et al., 2013*). This resource allows computational analysis of the entire organ in relation to cell staining at high resolutions (up to 20 µm). It is specially tailored for neuroimagers, as it is made available in common MRI formats (minc and NifTI), accompanied by cortical surface reconstructions (*Lewis et al., 2014*), and nonlinearly registered to standard MRI templates (ICBM152 and MNI-ADNI) (*Fonov et al., 2011a*). Furthermore, recent studies have expanded the resource by offering improved registrations to standard spaces (*Lewis et al., 2020*; *Xiao et al., 2019*), nuanced intracortical surface models, and laminar approximations (*Wagstyl et al., 2018a*; *Wagstyl et al., 2020*) as well as regional segmentations (*DeKraker et al., 2019*; *Xiao et al., 2019*). Several studies have already capitalised on this unique resource for integrative histological-neuroimaging analyses, including comparison of cytoarchitectural and functional gradients (*Paquola et al., 2019*), cross-validation of in vivo defined microstructural gradients in the insula with histological measures (*Royer et al., 2020*), mapping variations in functional connectivity along the histological axis of the mesio-temporal lobe (*Paquola et al., 2020b*), fMRI responses of the histologically defined auditory system (*Sitek et al., 2019*), comparison of cytoarchitectural similarity with MRI-derived estimates of structural connectivity (*Wei et al., 2019*), and analysis of the cytoarchitectural similarity of large-scale network hubs (*Arnatkevičiute et al., 2020*).

The present article introduces the BigBrainWarp toolbox. The aim of the toolbox is to facilitate integration of BigBrain with neuroimaging modalities, helping neuroscientists to utilise cytoarchitectural information in conjunction with in vivo imaging. The toolbox is open and includes (1) histological features and pre-transformed maps in BigBrain and imaging spaces, (2) code for performing data transformations, and (3) extensive tutorials. Toolbox functions and tutorials are documented on http://bigbrainwarp.readthedocs.io. Here, we introduce BigBrain to new users and demonstrate the utility of the BigBrainWarp toolbox. In the Materials and methods section, we overview the derivation of cytoarchitectural features from BigBrain and survey recent contributions to BigBrain-MRI integration. These include publication of histological cortical maps, regional segmentations, and registration efforts. Then, we detail the core functions of BigBrainWarp and the current contents of the toolbox. In the Results section, we share three tutorials to illustrate potential applications of BigBrainWarp.

## Materials and methods
### Overview of BigBrain

In brief, the reconstruction of BigBrain involved coronal slicing of a complete paraffin-embedded brain (65-year-old male) into 7404 sections at 20 µm thickness. Each section was stained for cell bodies (*Merker, 1983*), digitised, and subjected to manual and automatic artefact repair. The digitised sections were reconstructed into a contiguous 3D volume. The volumetric reconstruction is available

**Table 1.** Surface constructions for BigBrain.

| Surfaces | Utility | Reference |
| --- | --- | --- |
| Grey and white | Initialisation and visualisation | *Lewis et al., 2014* |
| Layer 1/2 and layer 4 | Boundary conditions | *Wagstyl et al., 2018a* |
| Equivolumetric | Staining intensity profiles | *Waehnert et al., 2014* |
| Deep learning laminar | Laminar thickness | *Wagstyl et al., 2020* |
| Hippocampal | Initialisation and visualisation | *DeKraker et al., 2019* |
| Mesiotemporal confluence | Initialisation and visualisation | *Paquola et al., 2020a* |

Note: Initialisation broadly refers to an input for feature generation, for example creation of staining intensity profiles or surface transformations.

online at 40 μm, 100 μm, 200 μm, 300 μm, 400 μm, and 1000 μm resolutions (http://bigbrainproject.org). The 40 μm version is released as 125 individual blocks corresponding to five subdivisions in the *x*, *y*, and *z* directions, with overlap. 100 -1000 μm resolution volumes are provided as single files. The Merker staining labels cell bodies, similar to Nissl staining, with a high contrast between black cell bodies on a light background (*Merker, 1983*). In the digitised images, darker colouring is represented by lower numbers (8bit graphics: $0–2^8$ = black-white). It is common practice to invert the values of the intensity, such that image intensity increases with staining intensity.

The grey and white matter boundaries of the cortical surface released in 2014 contain 163,842 vertices on each hemisphere, with vertices aligned between pial and white surfaces (*Lewis et al., 2014*). Surfaces were generated using a modified version of CIVET (*Kim et al., 2005*; *MacDonald et al., 2000*). Since then, a number of additional surface reconstructions have been published from which we may attain a range of metrics (*Table 1*).

## Staining intensity profiles and derived features

Sampling staining intensity from many cortical depths provides a profile of the cytoarchitecture, hereafter referred to as a staining intensity profile. This is achieved by constructing a set of surfaces within the cortex, then sampling intensity estimates at matched vertices across the surfaces. The current approach involves equivolumetric surface construction, whereby a set of intracortical surfaces are initialised at equidistant depths, then modulated by cortical curvature (*Waehnert et al., 2014*). This holds advantages for histological data because laminae vary in thickness depending on cortical folding (*Bok, 1929*). The procedure can be deployed using dedicated python scripts (*Wagstyl et al., 2018b*) and is implemented in the BigBrainWarp toolbox (**sample_intensity_profiles.sh**).

Smoothing can be employed in tangential and axial directions to ameliorate the effects of artefacts, blood vessels, and individual neuronal arrangement (*Wagstyl et al., 2018a*). Smoothing across depths is enacted for each profile independently. Here, we use an iterative piece-wise linear procedure that minimises curve shrinkage (*Taubin, 1995*). The degree of smoothing is modulated by the number of iterations. In contrast, surface-wise smoothing is performed at each depth independently and involves moving a Gaussian kernel across the surface mesh. We tested the impact of such pre-processing choices by generating the profiles with a range of parameters (number of surfaces: 50–100, iterations of depth-wise smoothing = 2–10, FWHM of surface-wise smoothing = 0–8) (*Appendix 1—figure 1A*). Then, we examined how these parameters affected the shape of the staining intensity profiles, based on the number of peaks in the profile, and the spatial autocorrelation of staining intensity profiles (*Appendix 1—figure 1B*). Spatial autocorrelation was calculated as the average product-moment correlation of staining intensity profiles at various distances along the BigBrain surface mesh (distances: 1–50 steps). In this case, the number of steps represents the shortest path along the surface mesh, treating the edges of the surface mesh as a graph. Increasing the number of surfaces beyond 50 did not impact the spatial autocorrelation and led to small increases in the number of peaks in intensity profiles (*Appendix 1—figure 1C*). Depth-wise smoothing did not impact either outcome measure. As could be expected, surface-wise smoothing substantially increased spatial auto-correlation. For the initial BigBrainWarp release, we selected 50 surfaces, 2 iterations of depth-wise

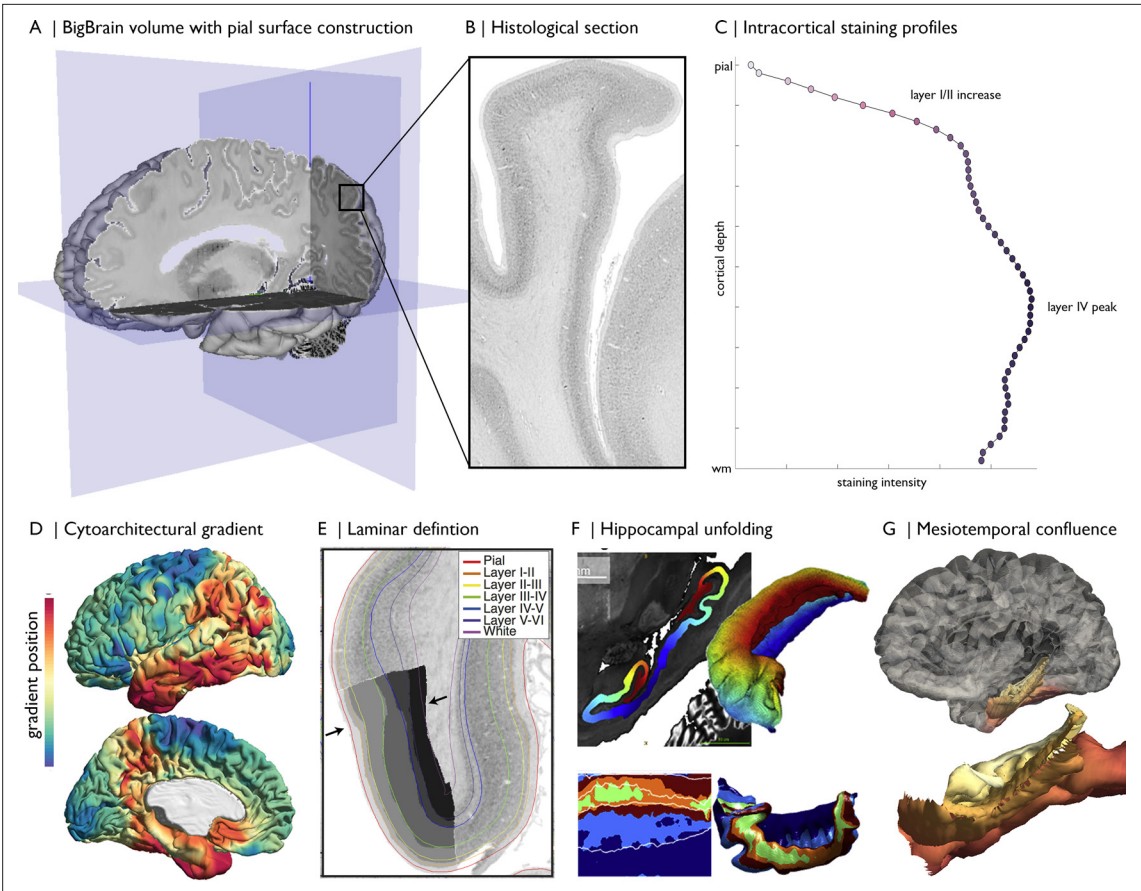

**Figure 1.** Magnification of cytoarchitecture using BigBrain, from (**A**) whole brain 3D reconstruction (taken on https://atlases.ebrains.eu/viewer) to (**B**) a histological section at 20 μm resolution (available from bigbrainproject.org) to (**C**) an intracortical staining profile. The profile represents variations in cellular density and size across cortical depths. Distinctive features of laminar architecture are often observable i.e., a layer IV peak. Note, the presented profile was subjected to smoothing as described in the following section. BigBrainWarp also supports integration of previous research on BigBrain including (**D–E**) cytoarchitectural and (**F–G**) morphological models (*DeKraker et al., 2019*; *Paquola et al., 2020a*; *Paquola et al., 2019*; *Wagstyl et al., 2020*).

smoothing and (a modest) 2 FWHM surface-wise smoothing. BigBrainWarp also provides a simple function for generating staining intensity profiles (**sample_intensity_profiles.sh**).

Previous research has sought to characterise the laminar structure of the cortex using BigBrain staining intensity profiles (*Paquola et al., 2019*; *Schleicher et al., 1999*; *Wagstyl et al., 2018a*; *Zilles et al., 2002*). The isocortex contains six layers (*Brodmann, 1909*), certain features of which manifest on BigBrain staining intensity profiles. The transition from layer I to II exhibits a sharp increase in staining because layer I is only sparsely populated with cells, while the outer granular layer II has a higher density. Layer IV harbours a second, noticeable peak in cell staining, corresponding to dense packing of granule cells. The peak of layer IV corresponds to the division between supragranular and infragranular layers, which have markedly different roles in neural communication (*Buffalo et al., 2011*; *Felleman and Van Essen, 1991*; *Rockland and Pandya, 1979*). The relative depth of layer IV is also potentially informative, likely related to the propensity for feedforward vs feedback communication (*Beul et al., 2017*; *Sanides, 1962*; *Wagstyl et al., 2018a*), though the demarcation of feedforward and feedback projections is more multifactorial and complex (*Rockland, 2015*). A six-layered decomposition of the BigBrain isocortex has also been produced by training a convolutional neural network on manual annotations in 51 regions, then extending the model to the whole isocortex (*Wagstyl et al., 2020*; *Figure 1E*). Laminar thickness estimates aligned with prior histological studies (*Von Economo and Koskinas, 1925*), while increasing overall spatial precision. There remains difficulty in extending these approaches to cortex without clear laminar differentiation, however (i.e., anterior insula, mesiotemporal lobe).

More detailed characterisation of cytoarchitecture is offered by moment-based parameterisation of staining intensity profiles. This technique, pioneered by the Jülich group (*Schleicher et al., 1999*; *Zilles et al., 2002*), involves calculating the central moments (i.e., mean, the center of gravity, standard deviation, skewness, and kurtosis) of each staining intensity profile and the derivative profile, resulting in a multidimensional feature vector for each cortical point. Each central moment may be interpreted in neurobiological terms (*Zilles et al., 2002*). For example, the mean has been related to overall cellular density (*Wree et al., 1982*). It is higher in the primary visual cortex than in Brodmann area 45 than in the primary motor cortex, Brodmann area 4. In contrast, skewness varies from sensory to limbic areas (i.e., sensory-fugal) and indexes the balance of cellular density in infra- vs supra-granular layers (*Paquola et al., 2020b*). Comparison of profiles can illuminate large-scale patterns of cortical organisation. Observer-independent discrimination of cortical areas can be accomplished by comparing moment-based feature vectors between neighbouring vertices (*Schleicher et al., 1999*). The areal boundaries are defined where the feature vector exhibits a sudden shift. Over the past 20 years, this procedure has been employed in 23 *post mortem* brains, including BigBrain, resulting in a 3D probabilistic atlas of the human brain (*Amunts et al., 2020*). While this work is based on a selection of histological sections of each brain, recent work investigates solutions for mapping each section in a stack with the help of deep learning, in order to produce gapless 3D maps at full detail (*Schiffer et al., 2020*) and ultimately obtain a dense mapping of the BigBrain model.

Cortex-wide cytoarchitectural similarity may also be estimated by cross-correlating staining intensity profiles between different cortical locations (*Paquola et al., 2019*). We recently applied diffusion map embedding, a nonlinear manifold learning technique (*Coifman and Lafon, 2006*), to the profile cross-correlation matrix of BigBrain to identify principle axes of cytoarchitectural differentiation (*Paquola et al., 2019*; *Figure 1D*). Here, we replicated the approach with updated staining intensity profiles. Bearing in mind the high-dimensional matrix manipulation necessary for this procedure, we first decimated the BigBrain mesh from 327,684 to ~10,000 vertices. Mesh decimation involves selection of a subset of vertices that preserve the overall shape of the surface followed by retriangulation of the faces with only the selected vertices. We assigned non-selected vertices to the nearest selected vertex, based on shortest path on the mesh (ties were solved by shortest Euclidean distance). In this manner, all 327,684 vertices belong to one of ~10,000 parcels. Derivation of the cytoarchitectural gradients involved (1) averaging staining intensity profiles within each parcel, (2) pair-wise correlation of parcel-average staining intensity profiles (controlling for the global-average staining intensity profile), (3) transformation to a normalised angle matrix, and (4) diffusion map embedding of this matrix. Each eigenvector captures an axis of cytoarchitectural variation and is accompanied by an eigenvalue that approximates the variance explained by that eigenvector. Here, the first two eigenvectors explain approximately 42% and 35% of variance, respectively, and describe anterior–posterior and sensory-fugal axes (further details in **Tutorial 2**).

## Morphometric models in BigBrain

The high resolution of BigBrain allows for precise segmentation of anatomical structures. Manual segmentations of the putamen, caudate nucleus, globus pallidus pars externa, globus pallidus pars interna, nucleus accumbens, amygdala, thalamus, red nucleus, substantia nigra, subthalamic nucleus, and the hippocampus are available on Open Science Framework (https://osf.io/xkqb3/). Extending upon whole-structure segmentation, a recent study *DeKraker et al., 2019* used anatomical landmarks to create an internal coordinate system of the hippocampus. The approach involved solving Laplace's equation under three sets of boundary conditions: anterior–posterior, proximal–distal (relative to the subiculum), and inner–outer (*DeKraker et al., 2018*). Subsequently, the hippocampus can be 'unfolded', allowing examination of histological and morphometric features in a topologically continuous space (*Figure 1F*), in line with other surface-based studies of the hippocampus (*Bernhardt et al., 2016*; *Caldairou et al., 2016*; *Kim et al., 2014*; *Vos de Wael et al., 2018*). Furthermore, this 3D coordinate system enabled the creation of a continuous surface model of the mesiotemporal cortex (*Paquola et al., 2020b*). The hippocampus is typically excluded from cortical surface models due to its complex folding and unusual cytoarchitectural makeup, with Cornu Ammonis subfields being allocortical and the dentate gyrus an interlocked terminus. Using the proximal–distal axis of the hippocampus, we were able to bridge the isocortical and hippocampal surface models recapitulating the smooth confluence of cortical types in the mesiotemporal lobe, i.e. mesiotemporal confluence

(*Figure 1G*). The continuous surface model, defined by a pial/inner surface and a white/outer surface, can also be used to initialise equivolumetric surface constructions (*Waehnert et al., 2014*; *Wagstyl et al., 2018b*). We generated staining intensity profiles using 40 µm resolution blocks of BigBrain across the cortical confluence, which are released in BigBrainWarp with the matching surface model.

## BigBrain–MRI transformations

BigBrain–MRI integration is pillared upon transformations between spaces. Spatial registration already exists as a fundamental component of most neuroimaging pipelines. As such, extensive research has focused on the creation of standard spaces, such as ICBM-MNI152 (*Fonov et al., 2011b*; *Fonov et al., 2009*) and FreeSurfer's fsaverage (*Fischl et al., 1999*). Many studies have advanced registration techniques over the years (*Collins and Evans, 2011*; *Klein et al., 2009*; *Xiao et al., 2019*). Registration of BigBrain to MRI templates involves additional challenges, however, including histological artefacts, differences in intensity contrasts and inter-individual variability.

For the initial BigBrain release (*Amunts et al., 2013*), full BigBrain volumes were resampled to ICBM2009sym (a symmetric MNI152 template) and MNI-ADNI (an older adult T1-weighted template) (*Fonov et al., 2011a*). Registration of BigBrain to ICBM2009sym, known as BigBrainSym, involved a linear then a nonlinear transformation (available on ftp://bigbrain.loris.ca/BigBrainRelease.2015/). The nonlinear transformation was defined by a symmetric diffeomorphic optimiser (SyN algorithm; *Avants et al., 2008*) that maximised the cross-correlation of the BigBrain volume with inverted intensities and a population-averaged T1-weighted map in ICBM2009sym space. The Jacobian determinant of the deformation field illustrates the degree and direction of distortions on the BigBrain volume (*Figure 2Ai*, top).

A prior study (*Xiao et al., 2019*) was able to further improve the accuracy of the transformation for subcortical structures and the hippocampus using a two-stage multi-contrast registration. The first stage involved nonlinear registration of BigBrainSym to a PD25 T1-T2* fusion atlas (*Xiao et al., 2017*; *Xiao et al., 2015*), using manual segmentations of the basal ganglia, red nucleus, thalamus, amygdala, and hippocampus as additional shape priors. Notably, the PD25 T1-T2* fusion contrast is more similar to the BigBrainSym intensity contrast than a T1-weighted image. The second stage involved nonlinear registration of PD25 to ICBM2009sym and ICBM2009asym using multiple contrasts. The deformation fields were made available on Open Science Framework (https://osf.io/xkqb3/). The accuracy of the transformations was evaluated relative to overlap of region labels and alignment of anatomical fiducials (*Lau et al., 2019*). The two-stage procedure resulted in 0.86–0.97 DICE coefficients for region labels, improving upon direct overlap of BigBrainSym with ICBM2009sym (0.55–0.91 DICE) (*Figure 2Aii,Aiv*, top). Transformed anatomical fiducials exhibited 1.77 ± 1.25 mm errors, on par with direct overlap of BigBrainSym with ICBM2009sym (1.83 ± 1.47 mm) (*Figure 2Aiii,Aiv*, below). The maximum misregistration distance (BigBrainSym = 6.36 mm, Xiao = 5.29 mm) provides an approximation of the degree of uncertainty in the transformation. In line with this work, BigBrainWarp enables evaluation of novel deformation fields using anatomical fiducials and region labels ( **evaluate_warps.sh**). The script accepts a nonlinear transformation file for registration of BigBrainSym to ICBM2009sym, or vice versa, and returns the Jacobian map, Dice coefficients for labelled regions and landmark misregistration distances for the anatomical fiducials.

The unique morphology of BigBrain also presents challenges for surface-based transformations. Idiosyncratic gyrification of certain regions of BigBrain, especially the anterior cingulate, cause misregistration (*Lewis et al., 2020*). Additionally, the areal midline representation of BigBrain, following inflation to a sphere, is disproportionately smaller than standard surface templates, which is related to differences in surface area, in hemisphere separation methods, and in tessellation methods. To overcome these issues, ongoing work (*Lewis et al., 2020*) combines a specialised BigBrain surface mesh with multimodal surface matching (MSM; *Robinson et al., 2018*; *Robinson et al., 2014*) to co-register BigBrain to standard surface templates. In the first step, the BigBrain surface meshes were re-tessellated as unstructured meshes with variable vertex density (*Möbius and Kobbelt, 2010*) to be more compatible with FreeSurfer generated meshes. Then, coarse-to-fine MSM registration was applied in three stages. An affine rotation was applied to the BigBrain sphere, with an additional 'nudge' based on an anterior cingulate landmark. Next, nonlinear/discrete alignment using sulcal depth maps (emphasising global scale, *Figure 2Biii*), followed by nonlinear/discrete alignment using curvature maps (emphasising finer detail, *Figure 2Biii*). The higher-order MSM procedure that was implemented

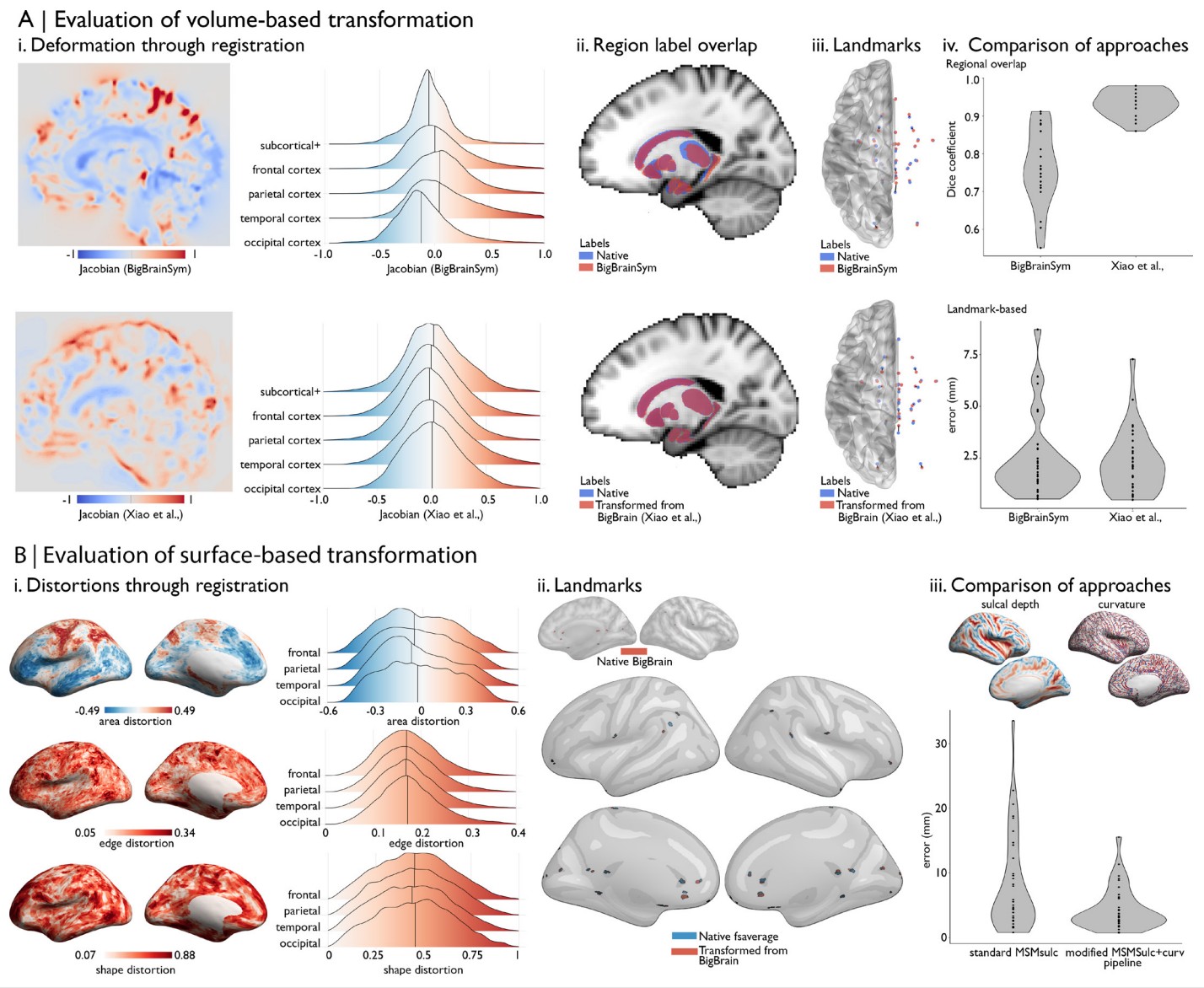

**Figure 2.** Evaluating BigBrain–MRI transformations. (**A**) Volume-based transformations. (**i**) Jacobian determinant of deformation field shown with a sagittal slice and stratified by lobe. Subcortical+ includes the shape priors (as described in Materials and methods) and the+ connotes hippocampus, which is allocortical. Lobe labels were defined based on assignment of CerebrA atlas labels (*Manera et al., 2020*) to each lobe. (**ii**) Sagittal slices illustrate the overlap of native ICBM2009b and transformed subcortical+ labels. (**iii**) Superior view of anatomical fiducials (*Lau et al., 2019*). (**iv**) Violin plots show the Dice coefficient of regional overlap (ii) and landmark misregistration (iii) for the BigBrainSym and Xiao et al., approaches. Higher Dice coefficients shown improved registration of subcortical+ regions with Xiao et al., while distributions of landmark misregistration indicate similar performance for alignment of anatomical fiducials. (**B**) Surface-based transformations. (**i**) Inflated BigBrain surface projections and ridgeplots illustrate regional variation in the distortions of the mesh invoked by the modified MSMsulc+ curv pipeline. (**ii**) Eighteen anatomical landmarks shown on the inflated BigBrain surface (above) and inflated fsaverage (below). BigBrain landmarks were transformed to fsaverage using the modified MSMsulc+ curv pipeline. Accuracy of the transformation was calculated on fsaverage as the geodesic distance between landmarks transformed from BigBrain and the native fsaverage landmarks. (**iii**) Sulcal depth and curvature maps are shown on inflated BigBrain surface. Violin plots show the improved accuracy of the transformation using the modified MSMsulc+ curv pipeline, compared to a standard MSMsulc approach.

for BigBrain maximises concordance of these features while minimising surface deformations in a physically plausible manner, accounting for size and shape distortions (*Figure 2Bi*; *Knutsen et al., 2010*; *Robinson et al., 2018*). This modified MSMsulc+curv pipeline improves the accuracy of transformed cortical maps (4.38 ± 3.25 mm), compared to a standard MSMsulc approach (8.02 ± 7.53 mm) (*Figure 2Bii–iii*; *Lewis et al., 2020*).

**Table 2.** Input parameters for the bigbrainwarp function.

| Parameter | Description | Conditions | Options |
|---|---|---|---|
| in_space | Space of input data | Required | bigbrain, bigbrainsym, icbm, fsaverage, fs_LR |
| out_space | Space of output data | Required | bigbrain, bigbrainsym, icbm, fsaverage, fs_LR |
| wd | Path to working directory | Required | |
| desc | Prefix for output files | Required | |
| in_vol | Full path to input data, whole brain volume. | | Permitted formats: mnc, nii or nii.gz |
| ih_lh | Full path to input data, left hemisphere surface | Requires either in_vol, or in_lh and in_rh | |
| ih_rh | Full path to input data, right hemisphere surface | | Permitted formats: label.gii, annot, shape.gii, curv or txt |
| interp | Interpolation method | Required for in_vol. Optional for txt input. Not permitted for other surface inputs. | For in_vol, can be trilinear (default), tricubic, nearest or sinc. For txt, can be linear or nearest |
| out_type | Specifies whether output in surface or volume space | Optional function for bigbrain, bigbrainsym and icbm output. Defaults to the same type as the input. | surface, volume |
| out_res | Resolution of output volume | Optional where out_type is volume. Default is 1 | Value provided in mm |
| out_den | Density of output mesh | Optional where out_type is surface. Default is 164 | For fs_LR out_space, 164 or 32 |

Note: the options are subject to change as the toolbox is expanded. Updates will be posted on https://bigbrainwarp.readthedocs.io/en/latest/pages/updates.html.

## Compiling BigBrainWarp

For BigBrainWarp, a modular set of scripts maps between common BigBrain and MRI spaces. Users need only interact with the overarching bigbrainwarp function (see *Table 2* for full functionality). The package automatically pulls state-of-the-art deformation fields and selects the appropriate transformation procedure, based on user inputs to bigbrainwarp (*Figure 3*). The bigbrainwarp function allows input and output of data that is aligned to the BigBrain volume, BigBrainSym volume, ICBM152 2009b symmetric volume, BigBrain surface (synonymous with BigBrainSym surface), fsaverage or fs_LR (164 k and 32 k versions). The type (i.e. volume or surface) is determined based on the input data. For volumetric input, the function is agnostic to voxel size, assuming an isomorphic resampling relative to the standard templates. For surface-based input, the data must contain a value for each vertex. By wrapping multiple forms of transformations into a single bash script (*Figure 3B–C*), we aim to reduce the onus on the user to have experience in the various software packages that are required by different registration procedures (e.g. minc-tools, FSL, HCP-workbench). Furthermore, containerisation of BigBrainWarp via Docker allows users to interact with the scripts without installing dependencies. This procedure ensures flexibility with ongoing developments in the field and simplifies procedures for new users.

We used BigBrainWarp to map histological gradients, discussed above, to fsaverage, fs_LR and ICBM152. Conversely, we used BigBrainWarp to transform in vivo derived microstructural and functional gradients, as well as intrinsic functional communities (*Yeo et al., 2011*), to the BigBrain surface. For the initial release of BigBrainWarp, we selected the multi-scale imaging connectomics (MICs) dataset, which contains group-level features on standard surface templates from 50 healthy adults (*Royer et al., 2021*). In particular, we adopted cortical gradients derived from qT1 mapping and resting-state functional connectivity. The current contents of the toolbox are shown in *Table 3*.

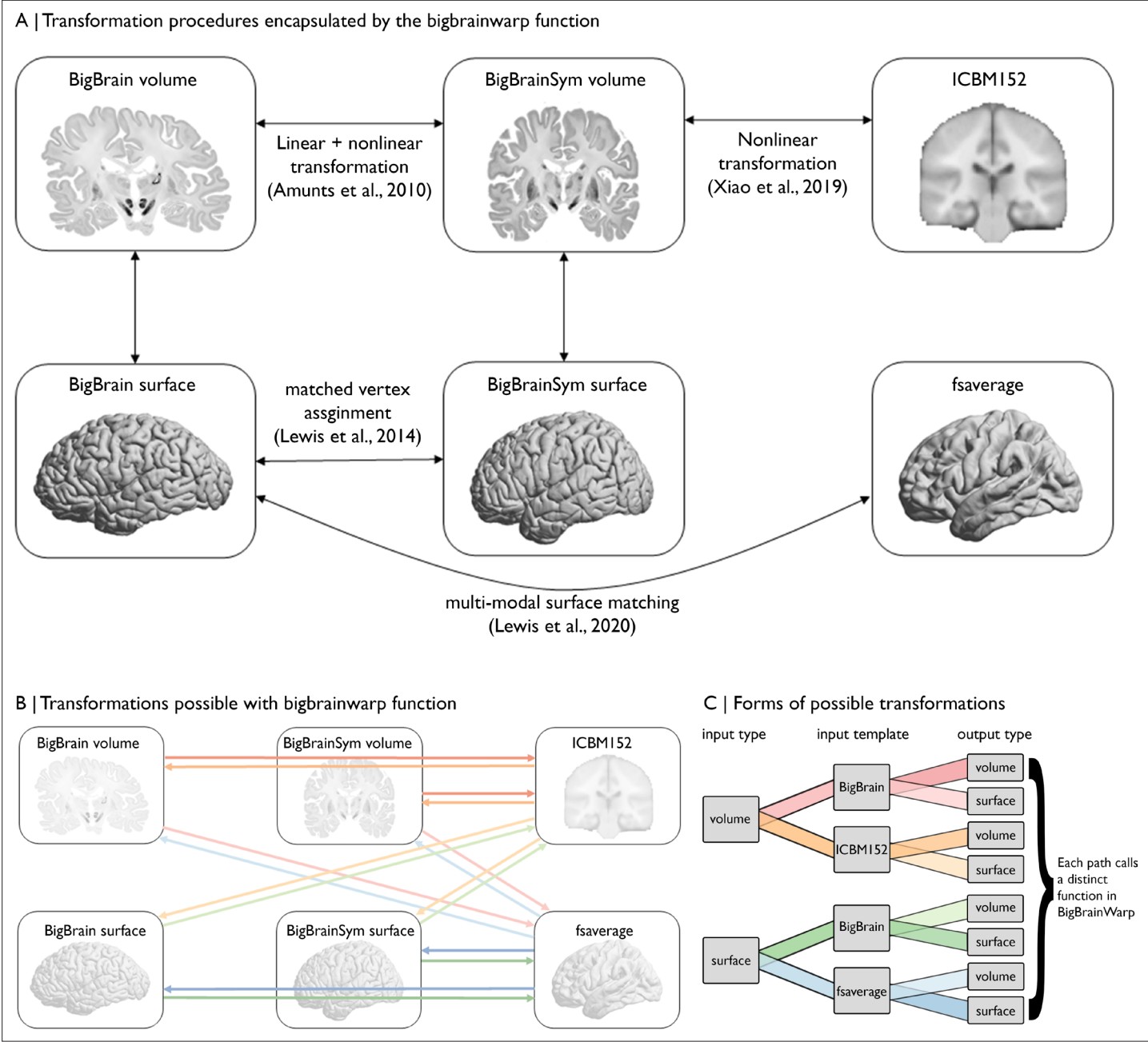

**Figure 3.** Overview of spaces and transformations included within BigBrainWarp. (**A**) The flow chart illustrates the extant transformation procedures that are wrapped in by the bigbrainwarp function. (**B**) Arrows indicate the transformations possible using the bigbrainwarp function. The colours, matched to C, reflect distinct functions called within BigBrainWarp. (**C**) The combination of input type, input template, and output type determines the function called by BigBrainWarp.

## Results

The BigBrainWarp toolbox supports a range of integrative BigBrain–MRI analyses. The following tutorials outline three BigBrain–MRI analyses with unique types of transformations, specifically (1) BigBrain volume to ICBM2009sym, (2) BigBrain surface to fsaverage, and (3) fsaverage to BigBrain surface. Neither the forms nor the motivations are exhaustive but illustrate applications (see *Figure 3* for all possible transformations). Code for each tutorial is available in the BigBrainWarp toolbox.

**Table 3.** BigBrainWarp contents.

| Data | Definition | Original space | Transformed spaces |
|---|---|---|---|
| Profiles | Staining intensity profiles, sampled at each vertex and across 50 equivolumetric surfaces | BigBrain | fsaverage, fs_LR (164 k and 32 k) |
| White | Grey/white matter boundary | BigBrain, fsaverage, fs_LR | |
| Sphere | Spherical representation of surface mesh | BigBrain, fsaverage, fs_LR | |
| Confluence | Continuous surface that includes isocortex and allocortex (hippocampus) from *Paquola et al., 2020a* | BigBrain | |
| Histological gradients | First two eigenvectors of cytoarchitectural differentiation derived from BigBrain | BigBrain | fsaverage, fs_LR (164 k and 32 k), icbm |
| Microstructural gradients | First two eigenvector of microstructural differentiation derived from quantitative in-vivo T1 imaging | fsaverage | BigBrain, |
| Functional gradients | First three eigenvectors of functional differentiation derived from rs-fMRI | fsaverage | BigBrain |
| Seven functional networks | Seven functional networks from *Yeo et al., 2011* | fsaverage | BigBrain |
| 17 Functional networks | 17 Functional networks from *Yeo et al., 2011* | fsaverage | BigBrain, icbm |
| Layer thickness | Layer thicknesses estimated from *Wagstyl et al., 2020* | BigBrain | fsaverage, fs_LR (164 k and 32 k) |

Note: Datasets Are Named According to BIDS and Align with Recommendations From TemplateFlow (*Ciric et al., 2021*).

## Tutorial 1: BigBrain → ICBM2009sym MNI152 space

Motivation: Despite MRI acquisitions at high and ultra-high fields reaching submillimeter resolutions with ongoing technical advances, certain brain structures and subregions remain difficult to identify (*Kulaga-Yoskovitz et al., 2015*; *Wisse et al., 2017*; *Yushkevich et al., 2015*). For example, there are challenges in reliably defining the subthalamic nucleus (not yet released for BigBrain) or hippocampal Cornu Ammonis subfields (manual segmentation available on BigBrain, https://osf.io/bqus3/; *DeKraker et al., 2019*). BigBrain-defined labels can be transformed to a standard imaging space for further investigation. Thus, this approach can support exploration of the functional architecture of histologically defined regions of interest.

Approach: (1) Create volumetric label in BigBrain space. (2) Perform nonlinear transformation to ICBM2009sym space using BigBrainWarp. (3) Transform individual resting-state functional MRI data to ICBM2009sym MNI152 space. (4) Sample timeseries from labelled voxels in this standard space.

Example: The mesiotemporal lobe plays important roles in multiple cognitive processes (*Moscovitch et al., 2005*; *Squire et al., 2004*; *Vos de Wael et al., 2018*) and is affected by multiple neurological and neuropsychiatric conditions (*Ball et al., 1985*; *Bernhardt et al., 2016*; *Bernhardt et al., 2015*; *Calabresi et al., 2013*). Increasing research suggests that this region shows complex subregional structural and functional organisation. Here, we illustrate how we can track resting-state functional connectivity changes along the latero-medial axis of the mesiotemporal lobe, from parahippocampal isocortex toward hippocampal allocortex, hereafter referred to as the iso-to-allocortical axis. For further details and additional motivation, please see *Paquola et al., 2020a*: (1) Our volumetric label represents the iso-to-allocortical axis of the mesiotemporal lobe. We constructed this axis by joining the isocortical (*Lewis et al., 2014*) and hippocampal (*DeKraker et al., 2019*) surface meshes in BigBrain histological space, creating the mesiotemporal confluence (available in BigBrainWarp, *Figure 1G*). Then, we calculated the distance of each vertex in the new surface model to the intersection of isocortical and hippocampal meshes, defining the iso-to-allocortical axis (*Figure 4A*). Next, we filled voxels in cortical ribbon according to the position of the iso-to-allocortical axis, producing a

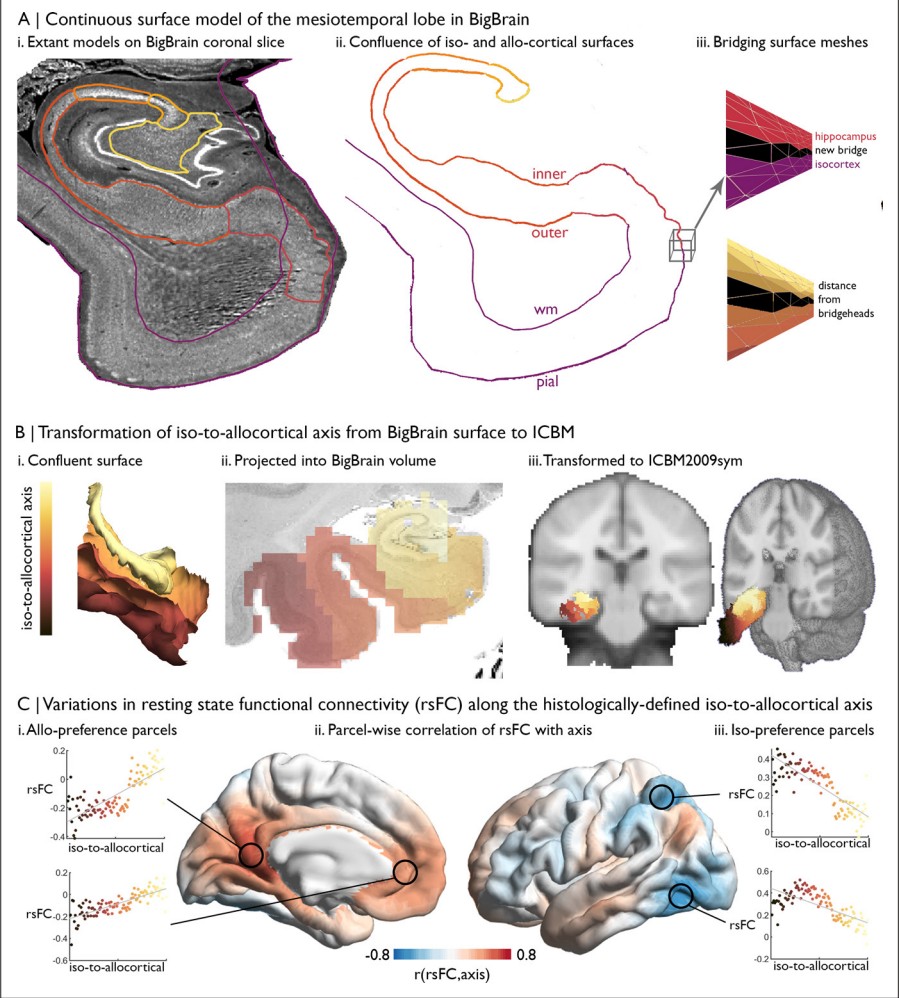

**Figure 4.** Intrinsic functional connectivity of the iso-to-allocortical axis of the mesiotemporal lobe. (**A**) i. BigBrain surface models of the isocortex and hippocampal subfields are projected on a 40 μm resolution coronal slice of BigBrain. (**ii–iii**) The continuous surface model bridges the inner hippocampal vertices with pial mesiotemporal vertices (entorhinal, parahippocampal or fusiform cortex). Vertices at the medial aspect of the subiculum were identified as bridgeheads and used to bridge between the two surface constructions. Geodesic distance from the nearest bridgehead was used as the iso-to-allocortical axis. (**B**) Iso-to-allocortical axis values were projected from the surface into the BigBrain volume, then transformed to ICBM2009sym using BigBrainWarp. (**C**) Intrinsic functional connectivity was calculated between each voxel of the iso-to-allocortical axis and 1000 isocortical parcels. For each parcel, we calculated the product-moment correlation (r) of rsFC strength with iso-to-allocortical axis position. Thus, positive values (red) indicates that rsFC of that isocortical parcel with the mesiotemporal lobe increases along the iso-to-allocortex axis, whereas negative values (blue) indicate decrease in rsFC along the iso-to-allocortex axis.

volumetric representation of the iso-to-allocortical axis in BigBrain histological space (*Figure 4Bii*). (2) We transform the volume from the BigBrain histological space to ICBM2009sym (*Figure 4Biii*).

```
bigbrainwarp --in_space bigbrain --out_space icbm --wd/project/ --desc
confluence_axis --in_vol tpl-bigbrain_desc-confluence_axis.nii --interp linear
```

(3) aboveTo explore the functional architecture of this histologically defined axis, we obtained multi-modal MRI in 50 healthy adults from the MICs dataset (*Royer et al., 2021*). For each participant, we constructed an individualised transformation from ICBM2009sym to their native functional space, based on the inverse of the within-subject co-registration to the native T1-weighted imaging concatenated to the nonlinear between-subject registration to ICBM2009sym. (4) For each participant, BOLD timeseries were extracted from non-zero voxels of the transformed iso-to-allocortical axis, which are classified as grey matter (>50% probability) and collated in a 3D matrix (voxel × time × subject). Then,

we sorted and analysed this matrix using the voxel-wise values of the iso-to-allocortical axis. For each subject, we averaged voxel-wise BOLD timeseries within 100 bins of the iso-to-allocortical axis and within 1000 isocortical parcels (*Schaefer et al., 2018*) and estimated the resting state functional connectivity of each iso-to-allocortical bin with each isocortical parcel. Then, we averaged resting state functional connectivity measures across subjects and performed product-moment correlations between the strength of resting state functional connectivity and bin position along the iso-to-allocortical axis. This analysis illustrates how functional connectivity varies along the histological axis for different areas of the isocortex (*Figure 4C*).

## Tutorial 2: BigBrain → fsaverage

Motivation: In vivo brain imaging reveals regionally variable effects of many demographic and clinical factors on brain structure and function. For example, prior studies of lifespan processes presented spatially variable patterns of cortical atrophy with advancing age, together with increased deposition of pathological aggregates, such as amyloid beta (Aβ) (*Bilgel et al., 2018*; *Jansen et al., 2015*; *Knopman et al., 2018*; *Rodrigue et al., 2012*; *Sperling et al., 2011*). Histological data provides a window into the cytoarchitectural features that align with imaging-derived phenotypes and that, in this instance, may predispose an area to specific aging-related processes. Essentially, we can evaluate whether regions with a certain cytoarchitecture overlap with those showing more marked aging effects. Furthermore, large-scale cytoarchitectural gradients can provide a unified framework to describe topographies, simplifying and standardising the reporting of imaging-derived phenotypes.

Approach: (1) Construct histological gradients using BigBrain and (2) transform to standard neuroimaging surface template using BigBrainWarp. (3) Plot the imaging-derived map against each histological gradient to understand the algebraic form of the relationship. Note, if imaging features are volumetric, one may use registration fusion to resample the data from ICBM2009sym to fsaverage (*Wu et al., 2018*). (4) Fit a statistical model to evaluate the relationship between the cytoarchitectural gradients and the imaging-derived map. For research questions with a more restricted region of interest, the cytoarchitectural gradient could be reconstructed within that field of view and the same procedure could be utilised. The optimal number of cytoarchitectural gradients should be evaluated.

Example: Cytoarchitectural correlates of age-related increases in Aβ deposition in a healthy lifespan cohort (*Lowe et al., 2019*; *Park, 2018*). (1–2) First, we obtained histological gradients on fsaverage from BigBrainWarp. The construction of histological gradients is detailed in Materials and methods (*Figure 5A*). The transformation from BigBrain to fsaverage was performed for the toolbox, like so,

```
bigbrainwarp --in_space bigbrain --out_space fsaverage --wd/project/
--desc Hist_G1 --in_lh Hist_G1_lh.txt --in_rh Hist_G1_rh.txt
```

For this analysis, we additionally smoothed the histological gradients on fsaverage (6 mm FWHM) to approximately match the smoothing kernel of the resting-state fMRI data. (3) We previously estimated the association of age with Aβ deposition across the cortical surface by combining positron emission tomography with MRI data in 102 adults (30–89 years), and assessed correspondence to functional connectivity gradients (*Lowe et al., 2019*). Here, we plot the vertex-wise t-statistics against Hist-G1 and Hist-G2 (*Figure 5B*). (4) We determine the optimal model via the Bayesian Information Criterion in univariate and multivariate regressions between the t-statistics and histological gradients (*Figure 5C*). The optimal model included only Hist-G2, indicating that Aβ preferentially accumulates towards the more agranular anchor of the sensory-fugal gradient.

## Tutorial 3: fsaverage → BigBrain

Motivation: A core aim of fMRI research is to map functional specialisation in the brain (*Bassett et al., 2008*; *Eickhoff et al., 2018*; *Gordon et al., 2017*; *Raichle, 2015*; *Shine et al., 2019*; *Yeo et al., 2011*). On the one hand, this work follows a long legacy of defining cortical areas, and on the other hand, it extends beyond the possibilities of *post mortem* research by capturing patterns of coordinated activity. For instance, clustering resting-state fMRI connectivity reveals a robust set of intrinsic functional networks (*Beckmann and Smith, 2004*; *Gordon et al., 2017*; *Yeo et al., 2011*). Nonetheless, there exists a gap in the literature between these well-characterised functional networks and their cytoarchitecture. BigBrain offers the opportunity to characterise and evaluate differences of cytoarchitecture for functionally defined atlases.

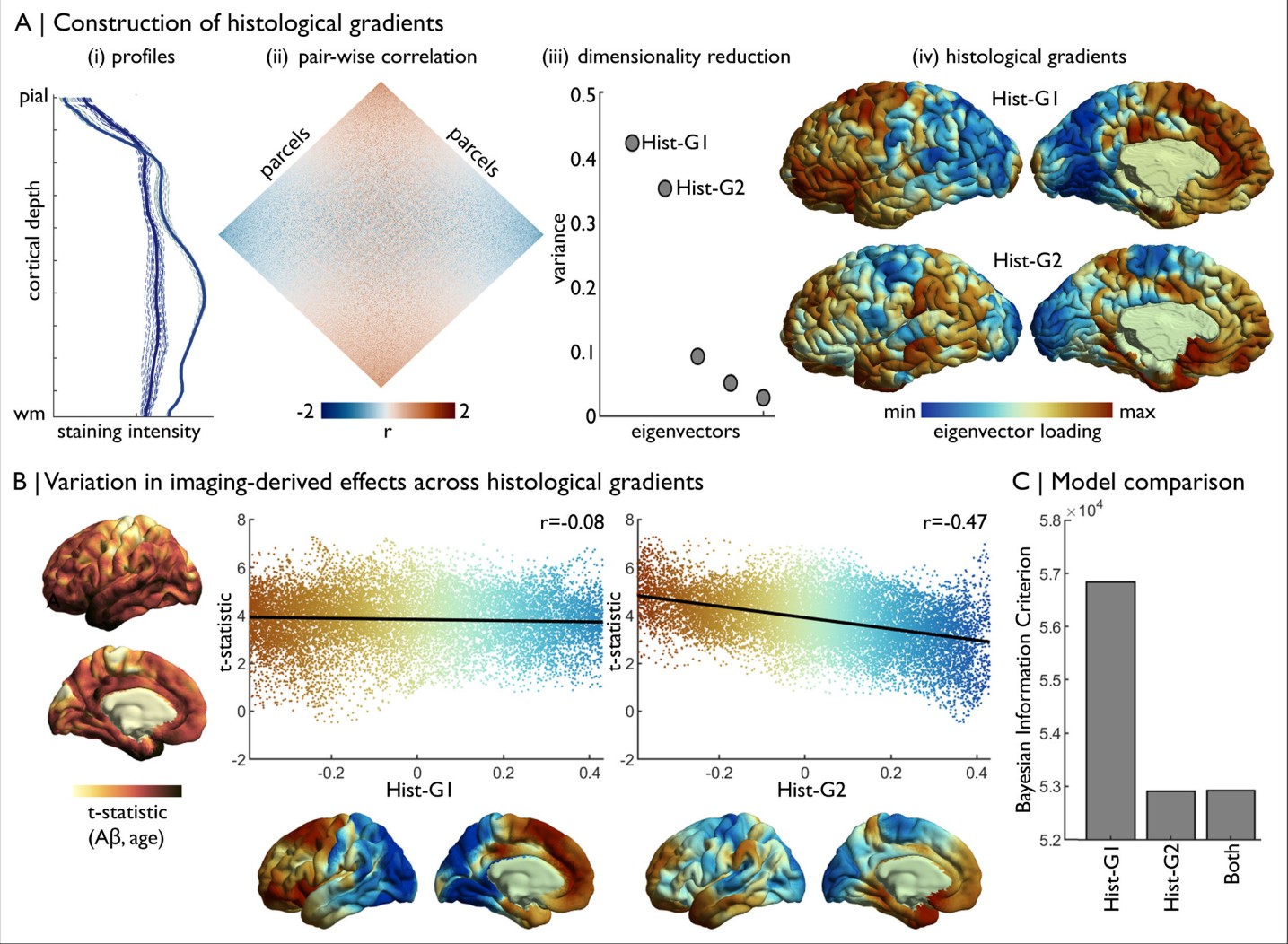

**Figure 5.** Concordance of imaging-derived effects with histological gradients. (**A**) Four stages of histological gradient construction. (i) Vertex-wise staining intensity profiles (dotted lines) are averaged within parcels (solid lines). Colours represent different parcels. (**ii**) Pair-wise partial correlation of parcel-average staining intensity profiles produces a cortex-wide matrix of cytoarchitectural similarity. (**iii**) The correlation matrix is subjected to dimensionality reduction, in this case diffusion map embedding, to extract the eigenvectors of cytoarchitectural variation. (**iv**) The eigenvectors capture histological gradients (Hist-G) and are projected onto the BigBrain cortical surface for inspection. (**B**) The t-statistic cortical map illustrates regional variations in the effect of age on Aβ deposition (*Lowe et al., 2019*), which was calculated vertex-wise on fsaverage5. To allow comparison, histological gradients were transformed to fsaverage5 using BigBrainWarp. Scatterplots show the association of the t-statistic map with the histological gradients. (**C**) Bar plot shows the Bayesian Information Criterion of univariate and multivariate regression models, using histological gradients to prediction regional variation in effect of age on Aβ deposition. The univariate Hist-G2 regression had the lowest Bayesian Information Criterion, representing the optimal model of those tested.

Approach: (1) Transform functionally-defined regions from a standard neuroimaging surface template to the BigBrain surface. Note, if the functional-defined regions are volumetric, one may use registration fusion to resample the data from ICBM2009sym to fsaverage (*Wu et al., 2018*). (2) Compile staining intensity profiles by functional class. (3) Assess discriminability of functional classes by staining intensity profiles.

Example: Cytoarchitectural differences of intrinsic functional networks. (1) Transform the 17-network functional atlas (*Yeo et al., 2011*) to the BigBrain surface.

```
bigbrainwarp --in_space fsaverage --out_space bigbrain --wd /project/
--desc Yeo2011_17Networks_1000 --in_lh lh.Yeo2011_17Networks_1000.annot
--in_rh lh.Yeo2011_17Networks_1000.annot
```

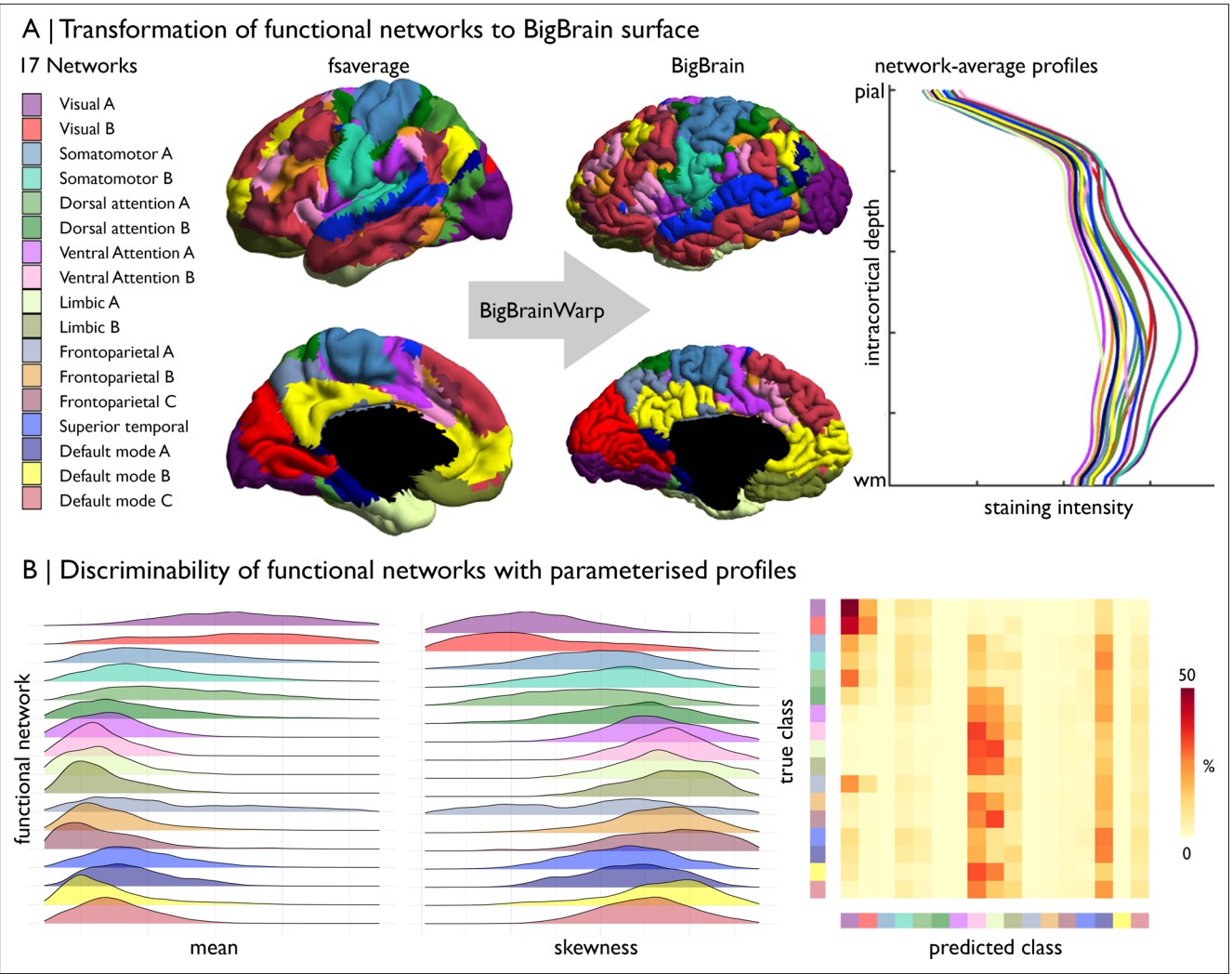

**Figure 6.** Prediction of functional network by cytoarchitecture. (**A**) Surface-based transformation of 17-network functional atlas to the BigBrain surface, operationalised with BigBrainWarp, allows staining intensity profiles to be stratified by functional network. (**B**) Ridgeplots show the moment-based parameterisation of staining intensity profiles within each functional network. The confusion matrix illustrates the outcome of mutli-class classification of the functional networks, using the central moment of the staining intensity profiles.

(2) Stratify staining intensity profiles by network (*Figure 6A*). (3) Parameterise staining intensity profiles by the central moments and assess variation across functional networks (*Figure 6B*). For example, the mean and skewness illustrate distinct patterns of cytoarchitectural differentiation across the functional networks. Visual networks have the highest mean and lowest skewness. Somatomotor, dorsal attention and fronto-parietal networks contain most variable mean and skewness values. Ventral attention, limbic, and fronto-parietal networks harbour the lowest mean and highest skewness, whereas the default mode networks occupy an intermediary position. Notably, all the networks exhibit broad distribution of the moments, signifying substantial cytoarchitectural heterogeneity, as well as overlapping values. To quantify discriminability of functional networks by cytoarchitecture, we can attempt to classify the functional networks using the central moments. For this example, we z-standardised the central moments and split the vertices into five folds, each with an equal representation of the 17 functional networks. Then, we trained a one vs one linear support vector classification on 50 % of each fold and tested the model on the remaining 50 % of that fold. Functional networks were equally stratified across training and testing. Finally, for each fold, we generated a confusion matrix, showing the accurate predictions on the diagonal and the incorrect classification off the diagonal. Predictive ability provides insight into distinctiveness and homogeneity of functional networks. Visual networks harbour distinctive cytoarchitecture, reflected by relatively high accuracy and few incorrect

predictions. Ventral attention, limbic, and temporo-parietal networks are relatively homogenous in cytoarchitecture, likely related to their restricted spatial distribution. The predictive accuracy did not appear to be negatively impacted by minor misalignments of the atlas, as the predictive accuracy was similar when excluding vertices within approximately 6 mm of the network boundaries (accuracy mean ± SD [%], original = 12.4 ± 15.4, excluding boundaries = 12.1 ± 13.3).

## Discussion

Beyond cartography, a major aim of neuroanatomical research has been to understand the functioning of the human brain. Throughout the 20th century, cytoarchitectural studies were instrumental in demonstrating functional specialisation across the cortex, as well as the uniqueness of the human brain amongst mammals (*Brodmann, 1909*; *Campbell et al., 1905*; *Sanides, 1962*; *Smith, 1907*; *Vogt and Vogt, 1919*; *Vogt, 1911*). Fine-grained anatomical resolution maintains an important role in understanding brain function in the modern era, helping to bridge between microcircuit organisation and macroscale findings obtained with in vivo neuroimaging. BigBrain is the first ultra-high-resolution 3D histological dataset that can be readily integrated with in vivo neuroimaging. In this report, we presented BigBrainWarp, a simple and accessible toolbox comprising histological data, previously developed transformation functions between BigBrain and standard imaging spaces, and ready-to-use transformed cortical maps. The toolbox is containerised to eliminate software dependencies and to ensure reproducibility. An expandable documentation is available, alongside several tutorials, at http://bigbrainwarp.readthedocs.io.

Multimodal registrations are core to integrating BigBrain with in vivo neuroimaging data. Identifying optimal solutions is more difficult than intra- and inter-subject co-registrations of neuroimaging data, owing to histological artefacts, differences in intensity contrasts and morphological distortions. These challenges have been addressed by recent studies, which improved integration of BigBrain with standardised MRI spaces. An automated repair algorithm was specially devised for BigBrain, which involved nonlinear alignment of neighbouring sections, intensity normalisation, outlier detection using block averaging then artefact repair using the block averages (*Lepage et al., 2010*; *Lewis et al., 2014*). Following initial transformation of BigBrain to ICBM2009b, which was part of the initial BigBrain release (*Amunts et al., 2013*), a recent study optimised subcortical registrations by generating a T1-T2* fusion contrast that is more similar to the BigBrain intensity contrast than a T1-weighted image (*Xiao et al., 2019*). Additionally, that study involved manual segmentation of subcortical nuclei to use as shape priors in the registration, which benefits the alignment of subcortical structures between BigBrain and standard neuroimaging templates. Finally, inspired by advances in the alignment of surface-based MRI data (*Robinson et al., 2018*; *Robinson et al., 2014*), the BigBrain team has recently developed a multi-modal surface matching pipeline for BigBrain that involved re-tessellation of the BigBrain surface at a higher resolution, followed by alignment to standard surface templates using coordinate, sulcal depth and curvature maps (*Lewis et al., 2020*). The procedure significantly improves upon previous techniques, resulting in geometric distortions comparable to those seen for registrations between neuroimaging datasets of different individuals (*Lewis et al., 2020*). Cortical folding is variably associated with cytoarchitecture, however. The correspondence of morphology with cytoarchitectonic boundaries is stronger in primary sensory than association cortex (*Fischl et al., 2008*; *Rajkowska and Goldman-Rakic, 1995a*; *Rajkowska and Goldman-Rakic, 1995b*). Incorporating more anatomical information in the alignment algorithm, such as intracortical myelin or connectivity, may benefit registration, as has been shown in neuroimaging (*Orasanu et al., 2016*; *Robinson et al., 2018*; *Tardif et al., 2015*). Overall, evaluating the accuracy of volume- and surface-based transformations is important for selecting the optimal procedure given a specific research question and to gauge the degree of uncertainty in a registration.

Practically, 3D histological models provide an unrivalled level of precision and lend novel opportunities to cross-validate and contextualise findings from human neuroimaging. BigBrainWarp is particularly well-suited for investigations on the fundamental relationships between cytoarchitecture and function, which remains an elusive aspect of brain organisation. Our tutorials illustrate a range of use cases of BigBrain-MRI integration. In tutorial 1, we show how BigBrain can be used to initialise region of interest analyses, such as mapping resting-state functional connectivity along the iso-to-allocortical axis (*Paquola et al., 2020b*), enabling delineation of regions that are difficult to identify with in vivo imaging and functional interrogation of histological axes. In tutorial 2, we show how cytoarchitectural

gradients can help to characterise large-scale cortical patterns, such as the association of aging with Aβ deposition (*Lowe et al., 2019*). This approach complements the tradition of reporting the cortical areas of significant clusters by offering a simplified topographical description of the spatial pattern. Furthermore, by comparing predictive power of various cytoarchitectural gradients, we may build towards hypotheses on the relationship between microcircuit properties and demographic or clinical factors. In tutorial 3, we discuss more specific histological features, namely moment-based parameterisation of staining intensity profiles (*Schleicher et al., 1999*; *Zilles et al., 2002*). These features depict the vast cytoarchitectural heterogeneity of the cortex and enable evaluation of homogeneity within imaging-based parcellations, for example macroscale functional communities (*Yeo et al., 2011*). The present analysis showed limited predictability of functional communities by cytoarchitectural profiles, even when accounting for uncertainty at the boundaries (*Gordon et al., 2016*). Together, these tutorials showcase how we can easily and robustly use BigBrain with BigBrainWarp to deepen our understanding of the human brain.

Despite all its promises, the singular nature of BigBrain currently prohibits replication and does not capture important inter-individual variation. While large-scale cytoarchitectural patterns are conserved across individuals, the position of areal boundaries relative to sulci vary, especially in association cortex (*Amunts et al., 2020*; *Fischl et al., 2008*; *Zilles and Amunts, 2013*). This can affect interpretation of BigBrain–MRI comparisons. For instance, in tutorial 3, low predictive accuracy of functional communities by cytoarchitecture may be attributable to the subject-specific topographies, which are well established in functional imaging (*Benkarim et al., 2020*; *Braga and Buckner, 2017*; *Gordon et al., 2017*; *Kong et al., 2019*). Future studies should consider the influence of inter-subject variability in concert with the precision of transformations, as these two elements of uncertainty can impact our interpretations, especially at higher granularity. Fortunately, the BigBrain team is working on new histology-based 3D models in the context of the HIBALL project (https://bigbrainproject.org/hiball.html). System neuroscience has dramatically benefitted from the availability of open resources (*Di Martino et al., 2014*; *Milham et al., 2018*; *Poldrack et al., 2017*; *Van Essen et al., 2013*). This path, together with ongoing refinements in multimodal data integration and efforts to make tools accessible, promises to further advance multi-scale neuroscience in the years to come.

## Acknowledgements

The project was conducted as part of the Helmholtz International BigBrain Analytics Learning Laboratory (HIBALL), an international initiative funded by Helmholtz Association & Healthy Brains for Healthy Lives. Casey Paquola was funded through the Fonds de la Recherche du Quebec – Santé (FRQ-S). Boris Bernhardt acknowledges research support from the National Science and Engineering Research Council of Canada (NSERC Discovery-1304413), the Canadian Institutes of Health Research (CIHR FDN-154298, PRJ-174995), SickKids Foundation (NI17-039), Azrieli Center for Autism Research (ACAR-TACC), BrainCanada (Azrieli Future Leaders), the Tier-2 Canada Research Chairs program and FRQ-S. Jessica Royer received support from a Canadian Institute of Health Research (CIHR) Fellowship. Ali Khan acknowledges research support from CIHR Project Grant #366062, NSERC Discovery Grant #6639, and the Canada First Research Excellence Fund.

## Additional information

### Funding

| Funder | Grant reference number | Author |
|---|---|---|
| Helmholtz Association | | Casey Paquola<br>Lindsay B Lewis<br>Claude Lepage<br>Jordan DeKraker<br>Katrin Amunts<br>Alan C Evans<br>Timo Dickscheid<br>Paule-J Toussaint<br>Sofie L Valk<br>Louis Collins<br>Boris Bernhardt |
| Fonds de Recherche du Québec - Santé | | Casey Paquola<br>Boris Bernhardt |
| Natural Sciences and Engineering Research Council of Canada | | Boris Bernhardt<br>Ali R Khan |
| Canadian Institutes of Health Research | | Jessica Royer<br>Ali R Khan<br>Boris Bernhardt |
| Sick Kids Foundation | | Boris Bernhardt |
| Azrieli Center for Autism Research | | Boris Bernhardt |

The funders had no role in study design, data collection and interpretation, or the decision to submit the work for publication.

### Author contributions

Casey Paquola, Conceptualization, Data curation, Formal analysis, Investigation, Methodology, Project administration, Software, Validation, Visualization, Writing – original draft, Writing – review and editing; Jessica Royer, Sofie L Valk, Validation, Writing – review and editing; Lindsay B Lewis, Claude Lepage, Data curation, Methodology, Writing – review and editing; Tristan Glatard, Software, Writing – review and editing; Konrad Wagstyl, Methodology, Software; Jordan DeKraker, Ali R Khan, Resources, Writing – review and editing; Paule-J Toussaint, Project administration, Writing – review and editing; Louis Collins, Methodology, Resources; Katrin Amunts, Conceptualization, Funding acquisition, Writing – review and editing; Alan C Evans, Conceptualization, Funding acquisition; Timo Dickscheid, Investigation, Methodology, Writing – review and editing; Boris Bernhardt, Conceptualization, Formal analysis, Investigation, Methodology, Writing – original draft

### Author ORCIDs

Casey Paquola ![iD] http://orcid.org/0000-0002-0190-4103
Paule-J Toussaint ![iD] http://orcid.org/0000-0002-7446-150X
Sofie L Valk ![iD] http://orcid.org/0000-0003-2998-6849
Louis Collins ![iD] http://orcid.org/0000-0002-8432-7021
Katrin Amunts ![iD] http://orcid.org/0000-0001-5828-0867
Boris Bernhardt ![iD] http://orcid.org/0000-0001-9256-6041

### Decision letter and Author response

Decision letter https://doi.org/10.7554/eLife.70119.sa1
Author response https://doi.org/10.7554/eLife.70119.sa2

## Additional files

### Supplementary files

• Transparent reporting form

## Data availability

All data generated or analysed during this study are included in the BigBrainWarp repository (https://github.com/caseypaquola/BigBrainWarp).

The following previously published datasets were used:

| Author(s) | Year | Dataset title | Dataset URL | Database and Identifier |
|---|---|---|---|---|
| Amunts K, Lepage C, Borgeat L, Mohlberg H, Dickscheid T, Rousseau M-E, Bludau S, Bazin P-L, Lewis LB, Oros-Peusquens A-M, Shah NJ, Lippert T, Zilles K, Evans AC | 2013 | BigBrain | https://bigbrain.loris.ca/main.php | LORIS, bigbrain |

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

# Appendix 1

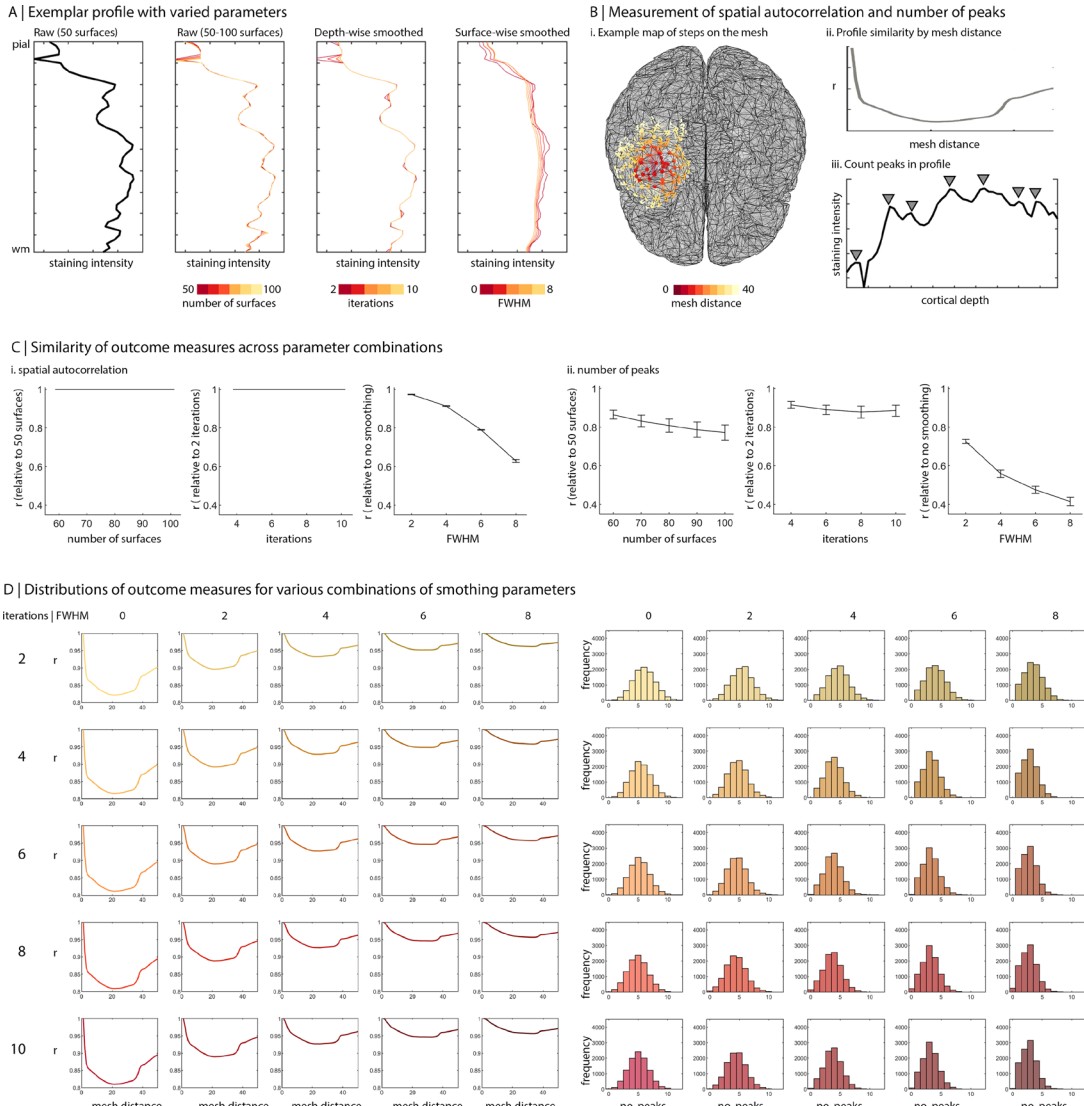

**Appendix 1—figure 1.** Influence of sampling parameters on staining intensity profiles. (**A**) Line plots show how the shape of an exemplar profile is changed by various sampling parameters. Far left is the raw profile constructed with 50 surfaces. Centre left are raw profiles constructed with 50–100 surfaces. Centre right are profiles (constructed with 50 surfaces) and subjected to varied levels of depth-wise smoothing. Far right are profiles (constructed with 50 surfaces and subjected to 10 iterations of depthwise smoothing) with varied levels of surface-wise smoothing. (**B**) Influence of sampling parameters was evaluated based on spatial autocorrelation and number of peaks. (**i–ii**) The spatial autocorrelation was defined by the number of steps between two vertices on the mesh, as depicted for an example vertex in (**i**). Then, we calculated the product-moment correlation between all staining intensity profiles and averaged these values based on the relative distance between vertices. The line plot show a decrease in correlation with increasing distance, attributable to spatial autocorrelation. (**iii**) The number of peaks was calculated to assess the jaggedness the staining intensity profile. (**C**) Using the lowest iteration of a sampling parameter as a baseline, we 31 calculated the product-moment correlation of profile features (spatial autocorrelation or number of peaks) with increases in the sampling parameter. In other words, the graph shows the similarity of solutions to the baseline sampling parameters. We found that the surface-wise smoothing impacts the spatial autocorrelation and number of peaks, while the number of surfaces and depthwise smoothing have little-to-no

*Appendix 1—figure 1 continued*

effect on spatial autocorrelation and a small effect on number of peaks. (**D**) For varying degrees of depth-wise (rows) and surface-wise (columns) smoothing, line plots show spatial autocorrelation and histograms show the distribution of number of peaks across profiles.

