## [Decision Letter]

**Acceptance summary:**

The manuscript introduces a new tool – BigBrainWarp – which consolidates several of the tools used to analyse BigBrain into a single, easy to use and well-documented tool. The BigBrain project produced the first open, high-resolution cell-scale histological atlas of a whole human brain. The tool presented here should make it easy for any researcher to use the wealth of information available in the BigBrain for the annotation of their own neuroimaging data. This is an important resource, with diverse tutorials demonstrating broad application.

**Decision letter after peer review:**

Thank you for submitting your article "BigBrainWarp: Toolbox for integration of BigBrain 3D histology with multimodal neuroimaging" for consideration by *eLife*. Your article has been reviewed by 3 peer reviewers, including Saad Jbabdi as Reviewing Editor and Reviewer #1, and the evaluation has been overseen by Tamar Makin as the Senior Editor. The following individuals involved in review of your submission have agreed to reveal their identity: Emma C Robinson, PhD (Reviewer #2); Roberto Toro (Reviewer #3).

Essential revisions: (see reviewer comments for details)

1) More in depth discussion of the limitations of having a single brain and the implications this has in utilising BigBrainWarp.

2) More details/clarifications on the registration approaches

*Reviewer #2 (Recommendations for the authors):*

I think this is a great resource and so I spent some time looking into the installation and running of the pipeline and in general I find the instructions very clear. This was even after having to install the pipeline from source as docker isn't working for me on MacOs Big Sur currently.

In terms of the paper specifically I had some initial trouble following the motivations, past research and tutorial experiments. As said previously the goals specific to this paper could be clearer in the abstract and Figure 1G is not clearly explained. For the Introduction, first paragraph, what is meant by ' the mergence of histology with computational neuroscience supports more observer independent principles?'

On 'staining profiles and derived features' I didn't follow the section on optimising smoothing and number of surfaces. What is meant by 'evaluated […] number of profile peaks for each combination' and comparing at 'various distances along the Bigbrain surface mesh' in terms of steps? I don't really find the supplemental figure on this helpful either – I can't interpret the grey bars in A.

In terms of the results, I think the examples used for the tutorials show broad applications and the detailed resource for running them (on the website) is great. I did have some trouble at first understanding what was done, however. So I recommend going back and writing in a more lay way. For example, tutorial (1) Figure 3C -its not immediately apparent whether this is from one subject or a combination. It took me some time to understand that the cortical surface colour map represents the trend from the plots of functional connectivity across the iso-to-allocortex axis. For tutorial 2 the text just says '(i) and (ii) are computed with BigBrainwarp' without stating what they are. The matrix in (ii) could have its dimensions labelled (i.e. regions x regions); and it isn't made clear (iii) is the eigenvalues and in the figure caption these are called 'principle components' but the decomposition is not PCA.

For tutorial 3 I think it would be wise to stress more carefully the impact of subject-specific topographic variation and the possible effect it has on the results – as mentioned in the public summary I think it may relate or interact with the findings of higher variation of the frontal parietal networks. Where topographic or cortical variation is mentioned it may be worth citing some papers with evidence for it e.g. HCP parcellation paper (Glasser Nature 2016) or Ruby Kong's 2021 Cerebral cortex paper, or Evan Gordon's parcellation papers.

*Reviewer #3 (Recommendations for the authors):*

The manuscript presents BigBrainWarp, a tool for facilitating the integration of BigBrain data for the analysis of neuroimaging data. The rationale for this is well described and the manuscript provides compelling illustrative applications.

---

## [Author Response]

Essential revisions: (see reviewer comments for details)1) More in depth discussion of the limitations of having a single brain and the implications this has in utilising BigBrainWarp.2) More details/clarifications on the registration approachesReviewer #2 (Recommendations for the authors):I think this is a great resource and so I spent some time looking into the installation and running of the pipeline and in general I find the instructions very clear. This was even after having to install the pipeline from source as docker isn't working for me on MacOs Big Sur currently.In terms of the paper specifically I had some initial trouble following the motivations, past research and tutorial experiments. As said previously the goals specific to this paper could be clearer in the abstract and Figure 1G is not clearly explained. For the Introduction, first paragraph, what is meant by ' the mergence of histology with computational neuroscience supports more observer independent principles?'On 'staining profiles and derived features' I didn't follow the section on optimising smoothing and number of surfaces. What is meant by 'evaluated […] number of profile peaks for each combination' and comparing at 'various distances along the Bigbrain surface mesh' in terms of steps? I don't really find the supplemental figure on this helpful either – I can't interpret the grey bars in A.

We edited this section in the Methods (P.5) for clarity and updated the *Figure* to more clearly demonstrate the evaluation criteria.

“Smoothing can be employed in tangential and axial directions to ameliorate the effects of artefacts, blood vessels, and individual neuronal arrangement (Wagstyl et al., 2018). […] As could be expected, surface-wise smoothing substantially increased spatial autocorrelation. For the initial BigBrainWarp release, we selected 50 surfaces, 2 iterations of depth-wise smoothing and (a modest) 2 FWHM surface-wise smoothing. BigBrainWarp also provides a simple function for generating staining intensity profiles (sample_intensity_profiles.sh).”

In terms of the results, I think the examples used for the tutorials show broad applications and the detailed resource for running them (on the website) is great. I did have some trouble at first understanding what was done, however. So I recommend going back and writing in a more lay way. For example, tutorial (1) Figure 3C -its not immediately apparent whether this is from one subject or a combination. It took me some time to understand that the cortical surface colour map represents the trend from the plots of functional connectivity across the iso-to-allocortex axis. For tutorial 2 the text just says '(i) and (ii) are computed with BigBrainwarp' without stating what they are. The matrix in (ii) could have its dimensions labelled (i.e. regions x regions); and it isn't made clear (iii) is the eigenvalues and in the figure caption these are called 'principle components' but the decomposition is not PCA.

We edited the Results (P.12, 14-15) for clarity, especially considering many readers are not familiar with the extant work that the tutorials draw upon.

“(iii) To explore the functional architecture of this histologically defined axis, we obtained multi-modal MRI in 50 healthy adults from the MICs dataset. For each participant, we constructed an individualised transformation from ICBM2009sym to their native functional space, based on the inverse of the within-subject co-registration to the native T1-weighted imaging concatenated to the nonlinear between-subject registration to ICBM2009sym. (iv) For each participant, BOLD timeseries were extracted from non-zero voxels of the transformed iso-to-allocortical axis, which are classified as grey matter (>50% probability) and collated in a 3D matrix (voxel ✕ time ✕ subject). Then, we sorted and analysed this matrix using the voxel-wise values of the iso-to-allocortical axis. For each subject, we averaged voxel-wise BOLD timeseries within 100 bins of the iso-to-allocortical axis and within 1000 isocortical parcels (Schaefer et al., 2018) and estimated the resting state functional connectivity of each iso-to-allocortical bin with each isocortical parcel. Then, we averaged resting state functional connectivity measures across subjects and performed product-moment correlations between the strength of resting state functional connectivity and bin position along the iso-to-allocortical axis. This analysis illustrates how functional connectivity varies along the histological axis for different areas of the isocortex (Figure 4C).

(i-ii) First, we obtained histological gradients on fsaverage from BigBrainWarp. The construction of histological gradients is detailed in the Methods (Figure 5A). The transformation from BigBrain to fsaverage was performed for the toolbox, like so,

bigbrainwarp --in_space bigbrain --out_space fsaverage --wd/project/ --desc Hist_G1 --in_lh Hist_G1_lh.txt --in_rh Hist_G1_rh.txt

For this analysis, we additionally smoothed the histological gradients on fsaverage (6mm FWHM) to approximately match the smoothing kernel of the resting state fMRI data.”

For tutorial 3 I think it would be wise to stress more carefully the impact of subject-specific topographic variation and the possible effect it has on the results – as mentioned in the public summary I think it may relate or interact with the findings of higher variation of the frontal parietal networks. Where topographic or cortical variation is mentioned it may be worth citing some papers with evidence for it e.g. HCP parcellation paper (Glasser Nature 2016) or Ruby Kong's 2021 Cerebral cortex paper, or Evan Gordon's parcellation papers.

We thank the Reviewer for their suggestions. We agree the interaction of subject-specific topographies and the accuracy of transformations requires special consideration in BigBrain-MRI integration. We expanded upon the issue in the revised *Discussion* (P.19)

“These features depict the vast cytoarchitectural heterogeneity of the cortex and enable evaluation of homogeneity within imaging-based parcellations, for example macroscale functional communities (Yeo et al., 2011). The present analysis showed limited predictability of functional communities by cytoarchitectural profiles, even when accounting for uncertainty at the boundaries (Gordon et al., 2016). […]Despite all its promises, the singular nature of BigBrain currently prohibits replication and does not capture important inter-individual variation at the scale of histology. For instance, in tutorial 3, low predictive accuracy of functional communities by cytoarchitecture may be attributable to the subject-specific topographies (Benkarim et al., 2020; Braga and Buckner, 2017; Gordon et al., 2017; Kong et al., 2019). Future studies should consider the influence of inter-subject variability and the precision of transformations, as these two elements of uncertainty jointly impact the accuracy of mapping between brains, especially with higher granularity parcellations.”